# Relevant Brachycera (Excluding Oestroidea) for Horses in Veterinary Medicine: A Systematic Review

**DOI:** 10.3390/pathogens12040568

**Published:** 2023-04-06

**Authors:** Vicky Frisch, Hans-Peter Fuehrer, Jessika-M. V. Cavalleri

**Affiliations:** 1Clinical Unit of Equine Internal Medicine, Department for Small Animals and Horses, University of Veterinary Medicine Vienna, Veterinärplatz 1, 1210 Vienna, Austria; 2Institute of Parasitology, Department of Pathobiology, University of Veterinary Medicine Vienna, Veterinärplatz 1, 1210 Vienna, Austria

**Keywords:** Diptera, Brachycera, fly, Equidae, equids, transmission, vector

## Abstract

In equine stables and their surroundings, a large number of insects are present that can be a nuisance to their equine hosts. Previous studies about dipterans transmitting infectious agents to Equidae have largely focused on Nematocera. For the preparation of this systematic review, the existing literature (until February 2022) was systematically screened for various infectious agents transmitted to Equidae via insects of the suborder Brachycera, including Tabanidae, Muscidae, Glossinidae and Hippoboscidae, acting as pests or potential vectors. The PRISMA statement 2020 (Preferred Reporting Items for Systematic Reviews and Meta-Analyses) guidelines for systematic reviews were followed. The two concepts, Brachycera and Equidae, were combined for the search that was carried out in three languages (English, German and French) using four different search engines. In total, 38 articles investigating Brachycera as vectors for viral, bacterial and parasitic infections or as pests of equids were identified. Only 7 of the 14 investigated pathogens in the 38 reports extracted from the literature were shown to be transmitted by Brachycera. This review clearly shows that further studies are needed to investigate the role of Brachycera as vectors for pathogens relevant to equine health.

## 1. Introduction

The order Diptera, uniting the “true flies”, is divided into two suborders, Nematocera and Brachycera, the latter being distinguished by shorter antennae [1]. The suborder Brachycera includes, amongst others, Tabanidae, Muscidae, Glossinidae and Hippoboscidae, which are considered of great veterinary medical importance [1,2].

Until now, a considerable volume of research regarding arthropod pests of animals has been published, but only a small number of studies focused on Brachycera as vectors transmitting infectious agents to Equidae. In this review, the literature concerning the suborder Brachycera was analyzed in regard to their potential as vectors for infectious agents transmitted to Equidae. The family Tabanidae comprises 13 different genera in Europe [3]. Tabanidae have been described as the prototype insect mechanical vector with no concurrent routes of transmission [4]. Mechanical transmission in contrast to biological transmission does not involve the development or replication of the pathogen in the flies, but rather it occurs exclusively via contamination or regurgitation [5,6]. Krinsky and colleagues described the adaptation of Tabanids to blood-feeding insects, thereby providing the opportunity to act as mechanical vectors. Anautogeny, telmophagy, large blood meals, long engorgement time and interrupted feeding are factors that increase the likelihood of pathogen transmission [7]. Further parameters such as blood meal residues and infectious agent content affect the ability of mechanical transmission [8,9]. In addition to these factors, the flies’ host preference, the pain perceived by the host, the host´s defense mechanisms and the distance to a new susceptible host all influence the potential risk of mechanical transmission of infectious agents [4,7].

Amongst others, the family Muscidae includes secretophagous non-biting flies such as *Musca domestica* (house fly), *Musca autumnalis* (face fly), and biting flies such as *Stomoxys calcitrans* (stable fly) are identified as pool feeders [10,11]. The potential of *Musca* spp. as mechanical and biological vectors derives from the contact of the fly with excretions, secretions or wounds of animals. The transfer successfully occurs by regurgitation of infectious agents through their sponging mouthparts, defecation of the fly or while in mechanical contact with the host due to phoresis of different infectious agents [11,12]. The potential of *Musca domestica* as a vector of disease is increased by their potential dispersal range of up to 12 km, the ability of both males and females to transmit infectious agents as well as by the stress caused by the persistent presence of multiple flies on the host [11,13]. The potential of *Stomoxys* spp. as mechanical vectors for infectious agents results from various adaptations of the flies to blood feeding including their piercing and sucking mouthparts. Transmission can potentially occur via regurgitation, defecation or phoresis of different infectious agents, traveling from one host to another via the flies´ body [6]. Both sexes of *Stomoxys* spp. need blood meals of approximately 12 µL of blood to reproduce [10]. Additionally, as for Tabanidae, the bites of *Stomoxys* spp. are painful and often interrupted due to the defensive responses of the host. Up to 1000 flies infesting one host have been reported and the dispersal range of flies has been documented for a distance of up to 225 km [10].

Glossinidae, a fly family well known as the tsetse fly, are the biological vectors of the protozoans causing African Trypanosomiosis. The flies play a role in the transmission of the protozoan parasite, as they are blood feeders (anautogeny) and support the cyclical development of the parasite. As soon as *Trypanosoma congolense*, *Trypanosoma brucei* spp. and *Trypanosoma vivax* enter the flies´ body, they develop and start to multiply either in the midgut or proboscis [14]. By biting the host, the flies transfer the infective trypanosomes into the skin [14]. The bite of Glossinidae has been described as a sharp prick [13,15,16]. In Africa, *Glossina morsitans*, *Glossina palpalis* and *Glossina fusca* are described as the most important Glossinidae that play the role of vectors [17].

Hippoboscidae, also known as louse flies, are bloodsucking flies. *Hippobosca equina* is the most common specie to parasitize Equidae [15,18]. Additionally, horses are discussed to be a facultative host of *Lipoptena cervi* [19].

Brachycera play a role as transmitting agents of various infectious diseases. One of the most well-studied pathogens transmitted by Brachycera is Equine Infectious Anaemia virus (EIAV), a retrovirus inducing persistent lifelong infection that can manifest as an acute, chronic or inapparent clinical disease. Equidae remain the only known family to be infected with EIAV. The virus can be found either free in the plasma, attached to circulating cells or residing within monocytes of infected equids [4,8]. The percutaneous introduction of infected blood has been described as the most common type of EIAV transmission and more precisely via Brachycera in a horizontal mechanical route of transmission [4,20]. Despite only a small number of species of Tabanidae that have been demonstrated as vectors for EIAV, they may constitute effective vectors [7]. Horses with pyrexia infected acutely with EIAV are suggested to be the best source of transmission, with viremias of up to ~10^6^ mL^−1^. However, inapparent carriers pose a risk as well [21]. In combination with up to 10^−6^ mL blood meal residues on the mouthparts of Tabanidae, possible transmission is considered realistic even with only partial deposition of the blood residues [22]. Younger horses are apparently more active in shooing away Tabanids, which could be one explanation for the higher infection rates of EIAV in adult horses [2]. Nevertheless, EIA is a lifelong disease and older equids have more time to simply be infected.

Viruses, of which transmission via Brachycera to Equidae is suspected, include African Horse Sickness virus, Vesicular Stomatitis virus and Bovine Papillomavirus, the latter of which has been associated with the development of equine sarcoids, a common skin tumor in Equidae [23,24,25,26,27]. Other viral infections affecting Equidae such as California Encephalitis virus, Western Equine Encephalitis virus, Tick-borne Encephalitis virus and Vesicular Stomatitis virus were detected in Tabanidae [5,7,28]. West Nile virus, a member of the Flaviviridae family, was detected in *S. calcitrans* during an outbreak of WNV infection in American white pelicans [29]. Since no viral replication occurred in stable flies, the biological transmission route was ruled out. *S. calcitrans* treading or sucking on swabs, contaminated with WNV-infected blood, successfully transmitted a viable virus to an uncontaminated swab [30]. Further, WNV viral RNA was detected by real-time RT-PCR in Hippoboscidae belonging to the ornithophilic species *Icosta americana* during an outbreak of WNV infection in captive owls and in injured or sick raptors [31,32]. However, the detection of WNV in these Brachycera provides information about the presence of this pathogen in a certain region but does not prove vector capability.

Not only can viral infections be transmitted, but bacterial infections transmitted via Brachycera have also been described. Bacteria discussed and reviewed as potentially transmitted by Tabanidae and infecting Equidae include *Bacillus anthracis*, *Borrelia burgdorferi*, *Brucella* spp., *Clostridium chauvoei*, *Coxiella brunetii*, *Erysipelothrix rhusiopathiae*, *Francisella tularensis*, *Listeria moncytogenes*, *Neorickettsia risticii* and *Pasteurella multocida*. *Bartonella* spp. are also associated with possible vector transmission by Tabanidae [5,7,33].

As reviewed, *Stomoxys* spp. are described as vectors potentially infecting Equidae with *Bacillus anthracis, Coxiella brunetii, Dermatophilus congolensis, Enterobacter sakzakii, Erysipelothrix rhusiopathiae, Francisella tularensis* and *Pasteurella multocida* [6,34]. *Bartonella* spp. are also suspected of being transmitted via stable flies [33]. It was demonstrated that *S. calcitrans* can harbor *Neorickettsia ristricii* but no attempt of transmission was successful [35].

*Musca autumnalis* was said to mechanically transmit *Brucella abortus* and *Trueperella pyogenes* and potentially cause infections in Equidae [12].

A series of potential equine pathogens were detected in *M. domestica* including *Acinetobacter* spp., *Aeromonas* spp., *Bacillus* spp., *Campylobacter* spp., *Clostridium* spp., *Coccobacillus* spp., *Enterobacter* spp., *Enterococcus* spp., *Escherichia* spp., *Klebsiella* spp., *Listeria* spp., *Pasteurella* spp., *Pseudomonas* spp., *Salmonella* spp., *Serratia* spp., *Shigella* spp., *Staphylococcus* spp. and *Streptococcus* spp., reviewed in [11]. Furthermore, *M. domestica* can harbor *C. pseudotuberculosis* for up to 24 h after 10 min of exposure to contaminated bovine blood agar, and cause a bacterial infection that can lead to ulcerative lymphangitis, guttural pouch empyema and both external and internal abscesses in horses [36,37,38].

*Hippobosca equina* and *L. cervi* are considered potential vectors for bacterial agents such as *Bartonella* spp. [18,19].

Detection and/or transmission of various parasite species affecting Equidae by Brachycera have been described [39]. *Trypanosoma evansi*, *Trypanosoma brucei*, *Trypanosoma congolense* and *Trypanosoma vivax*, protozoan parasites causing Surra (*T. evansi*) and Nagana (*Trypanosoma* spp.) often resulting in a fatal outcome in horses, are shown to be mechanically transmitted by biting flies including various species of the tribe Stomyxini [6,17,40,41,42,43,44]. Tabanidae were reported to be the most important mechanical vectors for *T. evansi* [5,7,41,44,45]. Additionally, *T. vivax*, *T. congolense* and *T. brucei* can be cyclically transmitted by Glossinidae. *T. vivax* was further shown to be mechanically transmitted in experimental studies by tsetse flies [5,14,17,40,46]. Hippoboscidae are also suspected to play a role in the transmission of *Trypanosoma* spp. [18,44].

Habronematidosis, a helminth infection of Equidae described in horses, donkeys and zebras, is induced by the two different genera *Habronema* and *Draschia,* including eight different species [47,48]. Among these, *Habronema microstoma* (syn. *Habronema majus*), *Habronema muscae* and *Draschia megastoma* are the best-researched species [47,49]. As reviewed, the parasite develops in Muscidae exiting as L3-stage larvae through its mouthparts. Deposited near mouthparts of Equidae, where the vector is feeding, the parasite is swallowed by the host and develops further in the stomach, leading to gastric habronematidosis. In cases when *Habronema* spp. are deposited near the eyes, the nostrils, muco-cutaneous junctions or wounds, ocular, pulmonary, muco-cutaneous or cutaneous Habronematidosis (summer sores), respectively, can develop as local inflammatory reactions [47,50,51,52].

Thelaziosis (Eyeworm infection), a helminth infection of the eye and associated tissues of equids with worldwide distribution, is caused by *Thelazia lacrymalis* and *Thelazia rhodesi* [53]. The course of the disease varies from asymptomatic to subclinical, or in cases where clinical signs developed, conjunctivitis, keratitis, corneal opacity and ulcers were described [53,54]. The infection of the Muscidae vector takes place during feeding on the lachrymal secretions of the host [53].

Parafilariosis, caused by *Parafilaria multipapillosa*, another helminthic disease of subcutaneous and intermuscular connective tissue, is suspected to be transmitted to horses by *Haematobia atripalpis* [15].

In addition to their vector potential, Brachycera are also considered to be equine pests. With their need for sucking blood leading to painful bites or feeding on secretions of their hosts, they induce stress [5,6,12]. During feeding, *M. autumnalis* mechanically damage the host´s eyes with their mouthparts, inducing defensive behavior in the affected vertebrates [12]. *Haematobia irritans* (Horn fly), a blood-feeding parasite of cattle, was found to play a role in equine ventral midline dermatitis. Horn flies accumulate on the shoulders, neck, withers and abdomen of horses [55,56]. Described as not the most likely vectors for infectious agents, the possibility of transmissions cannot be ruled out as these blood feeders aggregate around damaged skin areas, coming into contact with secretions [51]. Tabanidae have also reportedly induced reactions described as dermal nodules as a result of biting [55]. *Symphoromyia* spp., of the family Rhagionidae, a major biting pest of horses in Yellowstone National Park, were described as attacking horses severely in many high-elevation areas [57]. Other flies such as *Suragina* spp., as well as the exudate-feeding Chloropidae in the genera *Hippelates, Liohippelates* and *Siphunculina,* are discussed pests of horses [58,59]. While rasping the skin with spines on their legs to increase secretion production on the host´s skin, Chloropidae (eye gnats) are described to feed on equids besides *S. calcitrans.* Irritations of the eyes as well as the possibility of transmission of different bacterial and viral agents are linked to these flies [59].

## 2. Results

### 2.1. Results of the Literature Research

The PRISMA-guided literature research, presented in the PRISMA 2020 flow diagram (Figure 1), yielded 74,073 results (Pubmed 10,224, Web of Science 3918, Scopus 37,747, Cabi direct 22,184) [60]. Removal of the duplicates resulted in 20,645 remaining publications. More publications were removed based on their titles and whether or not the content could be determined by the abstracts. In total, 124 studies were sought for retrieval, as listed in Figure 1. The full texts of 96 reports could be retrieved and evaluated. During the screening of the references, in total, 44 titles were searched and full texts were retrieved. Hence, 140 publications were screened for eligibility. After the exclusion of 102 studies, 38 fit the inclusion criteria and were included in this review. Four different families of Brachycera are represented in the studies, including Glossinidae, Hippoboscidae, Muscidae and Tabanidae. The detection of pathogen DNA related to Equidae or events regarding transmission to Equidae were identified for 14 different infectious agents.

### 2.2. Results Concerning the Research Questions

#### 2.2.1. Role of Tabanidae in Transmission of Pathogens

##### Role of Tabanidae in Transmission of Viruses

The transmission routes of **Bovine Papillomavirus** are, as yet, not completely understood, though studies showing possible vector transmission of BPV 1 and 2 were described in horses and donkeys [24,61,62]. In an epidemiological study, DNA of BPV 1 and 2 were detected by highly sensitive E5 touch-down PCR in one thorax and two abdomens of *Tabanus sudeticus* (giant horse fly), but no virions were detected in the samples. The positive-tested flies were caught either on the prepuce of a sarcoid-affected donkey or nearby on another animal infected with the same BPV 1 variants, at the Rifugio degli Asinelli in Italy (Figure 2A,B) [62].

Biological vector transmission studies of **Equine Infectious Anaemia virus** via Tabanidae (Figure 3) were unsuccessful [63,64]. Mechanical transmission was shown by only one individual of *Tabanus fuscicostatus* and six single *Chrysops flavidus* (deer fly) with blood from horses with an acute EIAV infection [65,66]. Inapparent carriers also pose a risk of transmission, as demonstrated both in experimental and field studies [63,67,68]. A maximal time of 30 min has been demonstrated until now for transfer by horse fly between the infective source and the host [65]. The first species of Tabanidae suggested as possible vectors for EIAV included *Hybomitra frontalis* and *Tabanus sulcifrons* [64,69,70,85]. *Tabanus fuscicostatus*, *Tabanus quinquevittatus*, *Hybomitra lasiopthalma* and *Chrysops flavidus* were shown to be mechanical vectors for EIAV during experimental infection studies [63,65,66]. During a one-year-long experimental study by Foil et al. the transmission of EIAV appeared to occur in a seasonal pattern in southern Louisiana (Figure 2A) [67]. The horses became antibody positive from April until October, tested via an agar gel immunodiffusion test. In the same time period, first *Hybomitra lasiopthalma* and later *Tabanus lineola* were the most abundant Tabanidae, amongst others [67]. Throughout the studies, acutely infected horses showed higher body temperatures, fever, depression, loss of appetite, rapid weight loss, severe muscular weakness and even death [64,65,66,67,68,69,70]. Prophylactic measures described included isolation measures, control measures before and after the transfer of horses, culling of infected animals, elimination of the carcasses and fly control [69,70].

##### Role of Tabanidae in Transmission of Bacteria

Transmission of ***Bacillus anthracis*** to horses by *Tabanus rubidus* was experimentally achieved. Immediate transmission was seen in experiments with 1 to 80 Tabanidae transferred from infected hosts to naïve horses. The delayed (48 h) transfer of *B. anthracis* required at least 40 flies to infect a horse fatally and 10 flies to produce a mild course of the disease. Bacteremia was demonstrated microscopically prior to the death of the host [86].

##### Role of Tabanidae in Transmission of Parasites

Transmission of ***Trypanosoma evansi*** to horses has been demonstrated. After interrupted feeding of *Tabanus striatus* on an infected horse, the flies were transferred to healthy horses in less than 3 min and allowed to finish their blood meal. The infected horses developed pyrexia of up to 41.1 °C. Infection was evident in peripheral blood microscopically [87].

***Trypanosoma theileri*** DNA with high sequence similarity in a horse and in *Tabanus* sp. was first detected in Malaysia (Figure 2A) during a survey on trypanosomiasis in which no morphological identification of the fly was described. This finding suggests interspecies transmission. Pale mucous membranes were the only clinical sign the horse showed [71].

#### 2.2.2. Role of Muscidae in Transmission of Pathogens and as Pests

##### Role of Muscidae in Transmission of Viruses

In 1910, the transmission of **African Horse Sickness virus** by *S. calcitrans* (Figure 4 and Figure 5) to a healthy horse was described. The horse underwent two experimental infections with 30–40 bites from flies inoculated by biting horses with acute clinical signs immediately prior to the transmission attempt. The authors stated that the experiment proves successful transmission as the experimental horse died of typical AHS and all other possible means of transmission were excluded [88]. However, it remains very unlikely that *S. calcitrans* is a competent vector for this pathogen.

As described above, **Bovine Papillomavirus** transmission via vectors has been studied in different Brachycera spp. Bovine Papilloma viral DNA was detected by PCR in *Finnia carnicularis* (lesser house fly), *M. domestica* (Figure 6) during field studies and in *S. calcitrans* during field and experimental studies [24,25,61]. In the study conducted in Devon (UK) (Figure 2A), the flies harbored the same BPV1 DNA variant, which was detected on the sarcoid swabs from donkeys sampled at the same geographical location. Transmission of BPV is considered possible for short periods after exposure to BPV in papillomas or sarcoids. Fly exposure to bovine papillomas carries a higher risk of transmission than exposure to equine sarcoids [24].

*Stomoxys calcitrans*, like Tabanidae, has been investigated as a vector for **Equine Infectious Anaemia virus**. In older studies, describing the transmission of EIAV to horses via stable flies, the host was considered positive when showing clinical signs or by inoculating laboratory animals as no other laboratory methods were available [69,70,72,85]. Successful transmission of EIAV to ponies was shown with the Panama City strain of *S. calcitrans*. The transmission succeeded with at least 224 flies feeding first on an acutely infected pony before feeding on the host pony within approximately 10 min [89]. Later, EIAV transmission from a febrile-infected pony to another pony, once via 52 and once via 100 stable flies, was demonstrated [66]. Throughout these studies, acutely infected horses were described as showing higher body temperatures, fever, muscular weakness, rapid weight loss, marked depression, oedema or even death [66,69,70,72,89]. Prophylactic measures described in the literature coincide with those found for the prevention of transmission of EIAV by Tabanidae.

##### Role of Muscidae in Transmission of Bacteria

*Musca domestica*, *S. calcitrans* and *Haematobia irritans*, which were trapped on farms with infected horses, tested positive for ***Corynebacterium pseudotuberculosis*** DNA by real-time polymerase chain reaction-based fluorogenic 5’ nuclease assay. A positive correlation was shown between the incidence of infected horses and the percentage of flies that tested positive [36]. The mechanical transmission of *C. pseudotuberculosis* biovar *equi* via *M. domestica* (Figure 4) to healthy ponies was demonstrated [90]. Ponies exposed to infected flies showed similar clinical signs as the positive control ponies inoculated by applying infected swabs to the wounds. They showed classical pectoral swelling, to which the name “Pigeon Fever” refers, increased heat, sensitivity, purulent discharge, superficial abscesses and enlarged axillary lymph nodes. The horses showed significantly higher serological titers, as well as higher neutrophil counts and SAA. Positive bacterial cultures were obtained from the exposed ponies until the purulent discharge resolved between 7 to 13 days post-exposure. The mode of transmission of flies, for example, via regurgitation or contact, was not evaluated [90]. Authors suggest husbandry measures, including increased hygiene and insect control to reduce the potential risk of transmission and infection [36].

***Streptococcus equi* spp. *equi***, the infectious agent causing Strangles in equids, was detected in 0.54% of *M. autumnalis* (face fly) (Table 1) by qPCR. The face flies were collected on a thoroughbred farm in central California during an outbreak of *S. equi* spp. *equi* infection. The authors suggest instituting proper husbandry measures, biosecurity protocols and fly control as flies pose a potential risk of transmission [73].

##### Role of Muscidae in Transmission of Parasites

Of all the suspected dipterans, only the biological vectors *M. domestica* and *S. calcitrans* were proven to transmit ***Habronema muscae* and *Habronema microstoma***, respectively. A *Habronema*-specific polymerase chain reaction (PCR)-based assay revealed habronema DNA in the following three different body parts: head, thorax and abdomen [51,74]. In an experimental study, *Musca domestica* showed an infection rate of 4.54 L3 larvae of *H. muscae* per fly after being reared on infected faeces of a horse, whereby others reported up to 29 larvae per fly head [51,75,91]. The larvae move through the thoracic aorta of the fly and leave the proboscides by piercing the external membrane of the labella [91]. In the course of a study, conducted in India, screening Empididae (horse dung flies) for nematodes, *Habronema* were detected and considered potential vectors without describing the method of transmission [76]. The successful use of *M. domestica* and *S. calcitrans* for xeno-diagnosis for habronematidosis in horses was described [75,91]. During a study in Dubai (Figure 2B), 25.8% of *M. domestica* (Table 1) caught on a horse farm were positive for *H. muscae,* in contrast to *Stomoxys,* which tested negative. Moxidectin, a macrocyclic lactone, showed high efficacy against *H. muscae* infection. The horses tested negative via xenodiagnostics four weeks after treatment [75]. Seasonality was suspected as not being a risk factor for the infection of fly larvae with *Habronema* [74]. Horses infected with *H. muscae* showed granulomatous dermatitis as well as multiple abscesses in the lungs containing nematode larvae and eosinophilic inflammation [51].

In horses, infestations with ***Thelazia lacrymalis* and *Thelazia skrjabini*** have been reported [92]. Experimental studies indicated *M. autumnalis* as an intermediate host and competent vector in the transmission of the L3-stage larvae of *T. lacrymalis* to horses, as L3 was detected by microscopic examination in all body parts of the flies [77]. In western Massachusetts (Figure 2A), up to 3.6% of the face flies (Table 1) tested positive for *T. lacrymalis* and up to 26 larvae were detected in a single fly [77]. *Thelazia lacrymalis* larvae, which develop in face flies, were transferred to ponies, matured in the eyes that were exposed and could be recovered from the eyes. The nematodes were detected on the surface of the eyeball, under the eyelids, in the conjunctiva and nictitating membrane as well as in the lacrymalis glands, the excreting ducts and surrounding the eye [92,93]. *Thelazia lacrymalis* was recovered from male and female flies [92]. The transmission of *T. skrjabini* via *M. autumnalis* to horses can only be surmised [92].

Oocysts of ***Cryptosporidium parvum* and *Giardia lamblia*** were detected on the surface of Muscidae (not further defined) suggesting phoresis, possibly leading to mechanical transmission due to oocytes sticking onto the mouthparts and tarsal adhesive structures of the flies. Further, both parasites were detected in fly homogenates analyzed using fluorescent in situ hybridization (FISH) and immunofluorescent antibody (IFA) assays, showing a contamination inside the fly bodies. The muscids that tested positive were collected at four different locations in northwest Georgia (Figure 2A), one of which had horses housed at the site. At this last site of collection, the prevalence of *Giardia lamblia* detected in muscids was 8.33% and of *Cryptosporidium parvum* was 50% (Table 1). The horses were not tested for infection by the protozoans but the authors state that the flies undoubtedly play a role in the movement of these pathogens [78].

In notes published from 1904, *S. calcitrans* and *Musca brava* were mentioned as vectors of ***Trypanosoma*** between horses but a detailed description of the *Trypanosoma* species and the experimental studies was lacking [94]. It remains unclear if *M. brava* and *S. calcitrans* are competent vectors for these pathogens.

DNA of ***Theileria equi***, a haemotropic protozoan parasite causing equine piroplasmosis, was discovered in male and female *S. calcitrans* in three different locations in Hungary (Figure 2B), but the horses were not tested for equine piroplasmosis. A 100% sequence identity to earlier reported findings in Hungary, Ukraine and Switzerland was shown. The authors detected Theileriae only in the thorax and abdomen of stable flies and further interpreted an immediate mechanical transmission as less likely compared to delayed regurgitation [79].

##### Muscidae as Dermatological Pests

In an experimental study, equine dermatological problems such as crusts and ulcers described as ventral midline dermatitis resolved after fly control with fenvalerate against *H. irritans* in combination with medication against Onchocerciasis [80]. Treatment against the parasitic helminth alone was not sufficient. The fact that a reduction in fly numbers was also required for successful treatment implies that the horn fly plays a role in ventral midline dermatitis [80].

The preferred feeding site of *S. calcitrans* in large animals such as equids and cattle are the legs, so this is where blood sucking by the stable flies may lead to exudative dermatitis [81].

#### 2.2.3. Role of Glossinidae in Transmission of Pathogens

##### Role of Glossinidae in Transmission of Parasites

The first suspected transmission of ***Trypanosoma* sp**., at the time named “Haematozoon or Parasite of the Fly disease”, by *Glossina morsitans* to Equidae and other mammals was described. Seasonality was not thought to be a predisposing factor. The hosts showed weakness, loss of condition, pale mucous membranes, oedema, opacity of the cornea, a rise in body temperature and the parasites were detected microscopically in blood samples. At necropsy, abdominal and pericardial effusion, botfly larvae in the stomach, friable spleen and pale kidneys were described. At the time, grains of arsenic were prescribed as a therapeutic and prophylactic measure [16]. In north-western Ethiopia, the infection with *T. congolense* was found to be the most important trypanosomiosis in donkeys with *G. morsitans* as the only potential cyclical vector detected in this area. Lower body condition scores and lower mean packed cell volumes were significantly associated with infection in donkeys [82]. *Glossina pallidipes* was shown to be the only cyclical vector found in southern Ethiopia for *T. congolense*. A prevalence of 10.7% was detected for infections of donkeys living in areas where *G. pallidipes* was detected in traps. Infections with *T. congolense* were significantly associated with packed cell volumes below the reference range. In the area screened negative for tsetse flies, the donkeys were only infected with *T. vivax* [83].

#### 2.2.4. Role of Hippoboscidae in Transmission of Pathogens

##### Role of Hippoboscidae in Transmission of Bacteria

***Borrelia* sp.** DNA was detected by qPCR in 2.9% of *Hippobosca equina* flies (Figure 7, Table 1) collected from horses in Egypt (Figure 2B) but all of the blood samples of equids were tested negative by qPCR [84].

***Anaplasma* sp.** DNA was detected by qPCR in 3.8% of *Hippobosca equina* (Table 1) collected from horses in Cairo (Egypt) (Figure 2B) but was not further sequenced. All blood samples of horses on which the flies were collected were tested negative by qPCR. However, donkeys sampled in an area two hours’ drive away from Cairo were qPCR positive for *A. marginale* and *A. bovis* [84].

## 3. Discussion

In this literature review, 38 studies dealing with Brachycera as pests or vectors transmitting pathogens to Equidae were included. Equine Infectious Anaemia virus is the most researched pathogen transmitted by Brachycera to equids followed by pathogens causing Trypanosomiosis. Transmission of *T. vivax* and *T. congolense* to cattle or rats, respectively, by Tabanidae, is suspected, but there were no studies found in this literature search proving their transmission by Tabanidae to Equidae. Until now, only *T. evansi* has been proven to be transmitted to horses by *Tabanus striatus* [5,7,40,44,45]. An experimental study showing transmission of *Trypanosoma* sp. by *S. calcitrans* between rats after 72 h was suspected to be due to regurgitation [95]. Possible transmission, suspected via contamination of wounds, of *T. brucei* and *T. evansi* by *Musca* spp. was described [45]. In the case of Trypanosomiosis, no experimental studies proving transmission via Muscidae to Equidae were detected in the literature research. Hippoboscidae are suspected vectors, but transmission of these protozoa from Hippoboscidae to Equidae has not yet been demonstrated [45].

In the cases of California Encephalitis virus, Western Equine Encephalitis virus, Tick-borne Encephalitis virus, Vesicular Stomatitis virus and West Nile virus detected in Tabanidae, either Muscidae or Hippoboscidae, respectively, no experimental or natural transmission to Equidae studies were found in the literature search [5,7,26,28,29,30,31]. The same applies for bacterial pathogens detected on Brachycera, for which no transmission to Equidae study was detected and proven in studies, except for the mechanical transmission of *B. anthracis* and *C. pseudotuberculosis* [5,6,7,11,12,33,34,35]. Further research needs to be performed soon in order to evaluate transmission routes of pathogens by Brachycera to equids and to avoid spread in the near future.

Overall, transmission could not be demonstrated for every described infectious agent affecting equids. For seven of the reported pathogens, including AHSV, *B. anthracis*, *C. pseudotuberculosis*, EIAV, *Habronema* spp., *Thelazia lacrymalis* and *T. evansi*, experimental transmission events have been reported. EIAV was the most experimentally investigated pathogen [36,65,66,75,77,86,87,88,90]. In reports concerning *Anaplasma* sp., BPV, *Borrelia* sp., *Cryptosporidium*, *Giardia*, *S. equi*., *Thelazia skrjabini* and *Theileria equi*, the detection of pathogen DNA in the vector or oocysts on the vectors‘ surface have been studied but transmission events have not been proven as the equids at the same location were not tested or no viable infectious agents were detected [61,62,73,78,79,82,83,84]. BPV1/2 vector transmission by Brachycera has not been shown as no virions were detected in flies. The mechanical vector transmission adaptations of Tabanidae and Muscidae in combination with the detection of the same viral variants of BPV1/2 DNA in vectors and hosts implies a mechanical transmission [7,61,62,96]. The investigating notes on the role of *S. calcitrans* transmission of Trypanosomiosis cannot be seen as a confirmation of the vector potential, as the original studies mentioned could not be retrieved [94]. The transmission of Theileriae via regurgitation by *S. calcitrans* was described as more likely than mechanical transmission, although not much is known about the regurgitation process from the crop of *S. calcitrans* [79,97,98]. Additionally, as the disinfection process of the flies could have altered the results concerning the detection of the contaminated mouthparts with the pathogen, mechanical transmission cannot be excluded [79]. The demonstration of possible transmission of these pathogens should be investigated in the future both to raise awareness and to determine potential biohazards.

The studies concerning possible transmission of *Streptococcus equi* via *M. autumnalis* as well as the proof of transmission of *C. pseudotuberculosis* by Muscidae should increase awareness of the need for adequate isolation [36,73]. Isolation barns of infected patients in hospitals and in other facilities should include isolation from possible vectors including Brachycera in order to decrease the risk of possible transmission. If complete isolation is not possible due to lack of infrastructure, fly control should be performed and vector contamination should be decreased by, for example, bandaging exudative lesions.

In studies conducted at the beginning of the 20th century, the identification of the vector as well as of the pathogen, the host’s clinical signs, necropsy findings, therapies and prophylactic measures were not described in depth. Neither the morphological identification of *S. calcitrans*, nor the identification of AHS virus, nor detailed information regarding the clinical and pathological signs were provided in the studies [88]. In case of EIAV, transmission events were often described as positive when the infected host showed acute clinical signs such as elevated body temperature, as no laboratory methods for detection of the pathogen were available. Detailed descriptions of clinical signs, pathological findings and prophylactic measures are missing [69,70,72]. Transmission of *B. anthracis* to horses by Tabanidae was shown, but only confirmed by microscopic examination of the patients’ blood. No molecular biological diagnostic techniques were used [86]. Although transmission of AHSV, EIAV and *B. anthracis* was experimentally detected, only EIAV transmission was confirmed in more recent studies by serological testing of the host. Clear evidence is lacking of the demonstration of the transmission of AHSV and *B. anthracis* [66,86,88]. Therefore, results of those studies should be interpreted with caution. During the planning of future research projects, however, they should not to be neglected and scientists should bear them in mind, even though the standards were not the same as in today’s research.

For African Horse Sickness virus transmission, members of the nematoceran genus *Culicoides* were shown to be the most important vectors. The spread of AHSV by *S. calcitrans* and other biting insects is not considered epidemiologically important, in spite of the possibility of mechanical transmission [27,99]. No further details about studies proving these vectors as unimportant were detected in this literature search. Further research is needed to improve our knowledge about infections, vectors and routes of transmission. International travel is described as leading to the spread of disease either by introducing pathogens transmitted by local vectors or by importing new vectors [100]. Considering the large increase in international transportation of horses, inapparent carriers pose a high risk by introducing pathogens to regions considered pathogen free, for example, EIAV [101]. If in these cases transmission routes are not well studied, potential spread, e.g., via Brachycera, can be overlooked, leading to severe consequences.

Xenodiagnosis, a less invasive method for detection of pathogens compared to blood or secretion sampling, was successfully used in the detection of *H. muscae*. Further research is needed to develop standardized protocols of xenodiagnostics using Brachycera for the other pathogens described in this study in order to replace more invasive sampling methods. It could present an option as a screening method in various locations to identify local pathogens. Even if transmission by flies cannot be proven, the detection of pathogens on the vectors can give an overview of their presence. This method can be useful not only in veterinary medicine but also in human medicine. For example, cryptosporidiosis affecting horses seems often to present a subclinical course, though it is described as a zoonosis and, therefore, the transmission and spread is of interest to human medicine [102]. In every single case, the dispersal of the different Brachycera needs to be kept in mind in order to adequately assess the results, since a range of up to 225 km has been described [10].

Climate and climate change play a role in possible vector transmissions, as flight time of vectors is influenced by various weather conditions. Higher temperatures are described as prolonging the active lifetime and mobility [103]. These factors raise the potential of flying vectors to transmit pathogens, as with a prolonged flight time they can possibly head for new hosts stationed at a greater distance.

## 4. Materials and Methods

### 4.1. Literature Search

A systematic literature search was performed using the PRISMA statement 2020 (Preferred Reporting Items for Systematic Reviews and Meta-Analyses) guideline for systematic reviews [60]. The electronic search was conducted between the 12 and 20 of February 2022 using the four databases: Pubmed, Web of Science, Scopus and CAB direct. For the search, two different concepts were created, and the searches were each conducted in three languages: English, German and French. The first concept included words representing Equidae, with a focus on Equidae predominantly housed in Europe (Table 2). The second concept included search terms for Diptera with a focus on relevant Brachycera for Equidae, particularly in the category of genus and species found in Europe with the exception of Glossinidae (Table 3) [3,104,105,106,107,108,109,110].

To ensure the exclusion of duplicates, the citation program Citavi was used to create lists of the duplicates, which were double-checked by the author before being deleted to prevent errors.

### 4.2. Inclusion Criteria

The outcome of the literature search was screened for inclusion criteria by the first author. Selected content-related criteria were the investigation of Brachycera as vectors and pests, infectious agents transmitted by Brachycera and Equidae as hosts of infections transmitted by Brachycera. Studies published until the 12 February 2022 in English, German and French were included.

All titles were read and if those did not clearly demonstrate the eligibility for exclusion, abstracts were also studied. If the title referred obviously to another topic, for example, gynecological problems in mares, the publication was excluded by screening the title. If uncertainties occurred, consensus with the co-authors was reached. Experimental studies, epidemiological analysis, case reports, case series, congress presentations, reports of experimental studies and short communications were included. The reference lists of the eligible articles as well as of the reviews found during the literature search were also checked for relevant articles. Exclusion criteria included studies investigating Nematocera, Ixodida, animals other than Equidae, humans, reproduction, plants, laboratory methods and treatments of infectious agents not related to this search. Despite the inclusion in the literature search, studies concerning Oestroidea were excluded.

### 4.3. Data Extraction

For the review process and data extraction, the full texts were read by the first author. The relevant data of the eligible articles were extracted by answering the questions outlined in Table 4 by highlighting the answers in the text in Citavi, which were then automatically saved in a list related to the specific text in the project in the Citavi cloud. If the texts were too old and automatic transfer was not possible, the answers were manually added to the Citavi project.

## 5. Conclusions

Only half of the pathogens documented in the 38 studies extracted during the literature search proved to be transmitted by Brachycera. The other half were detected in the flies and so primarily only mechanical transmission can be assumed. In the search, studies on Brachycera were found showing the transmission of pathogens to other animals, or there was detection of pathogens on the flies only, but the transmission to Equidae was not investigated. The lack of knowledge about possible transmission routes of various infectious agents for equids poses a health risk to the equine population. In the future, further research is needed to investigate the role of Brachycera as vectors for pathogens transmitted to Equidae.

## Figures and Tables

**Figure 1 pathogens-12-00568-f001:**
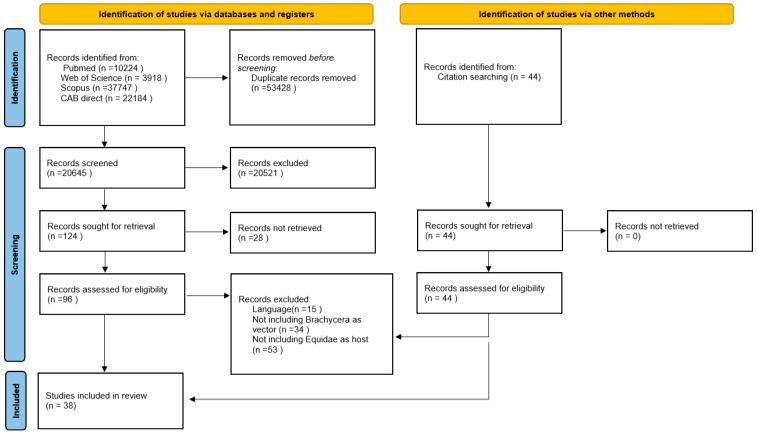
PRISMA 2020 flow diagram for new systematic reviews which include searches of databases, registers and other sources, visualizing the literature research.

**Figure 2 pathogens-12-00568-f002:**
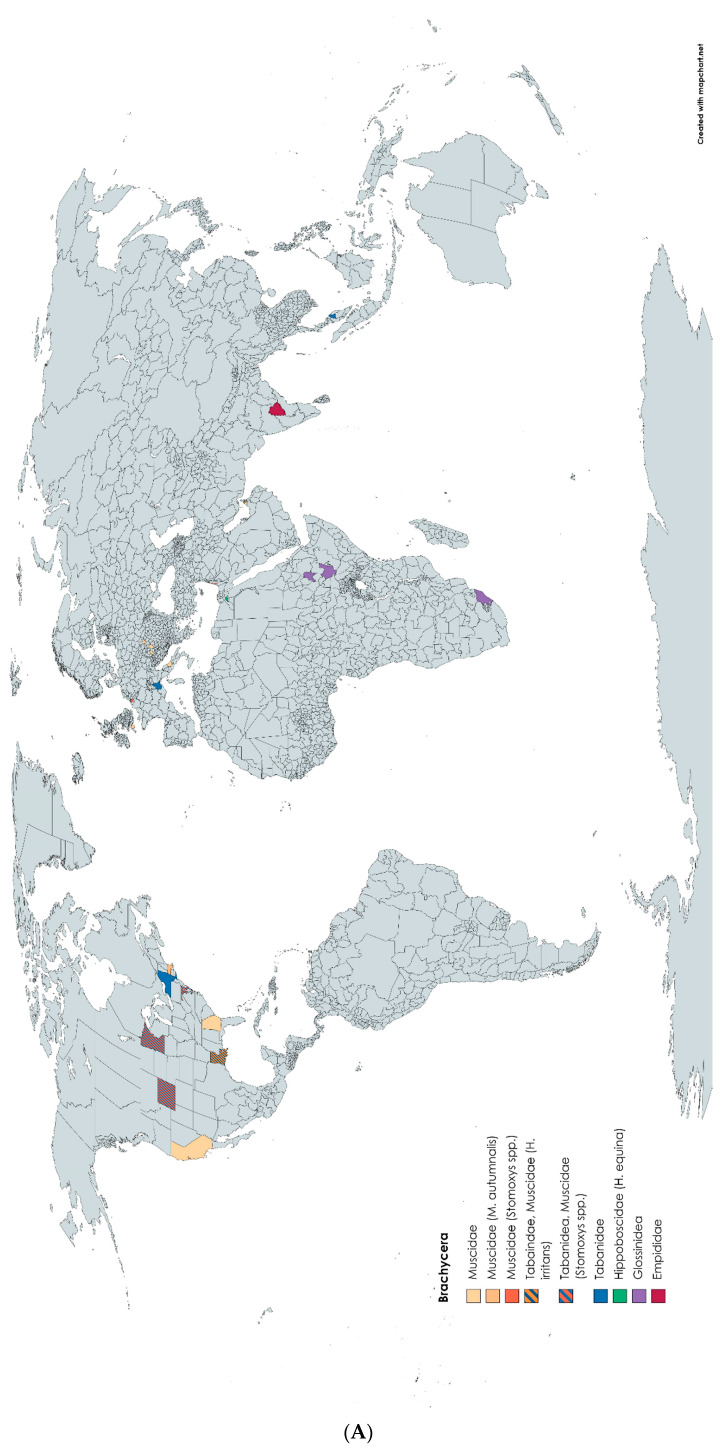
(**A**) Retrieved locations (regions) where studies on Brachycera in relation to their equine hosts were performed. Either flies were screened for pathogens, screened as pests for equids or transmission studies were executed [16,24,36,51,61,62,63,64,65,66,67,68,69,70,71,72,73,74,75,76,77,78,79,80,81,82,83,84] (transmission studies using laboratory bred flies were not included). (**B**) Enlarged window from Figure 7A. Retrieved locations (regions) in Europe and Middle East where studies on Brachycera in relation to their equine hosts were performed. Either flies were screened for pathogens, screened as pests for equids or transmission studies were executed [16,24,36,51,61,62,63,64,65,66,67,68,69,70,71,72,73,74,75,76,77,78,79,80,81,82,83,84] (transmission studies using laboratory bred flies were not included).

**Figure 3 pathogens-12-00568-f003:**
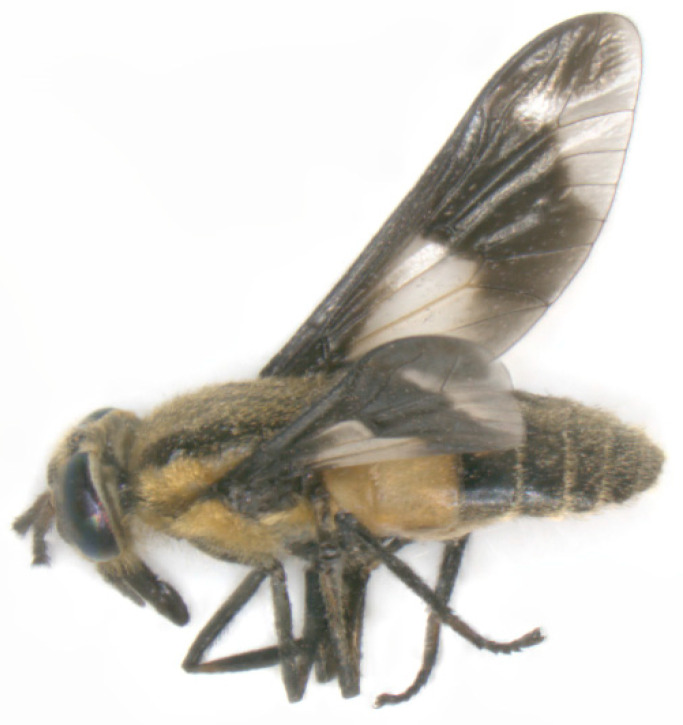
*Chrysops* sp. (9 to 10 mm body length) in lateral view.

**Figure 4 pathogens-12-00568-f004:**
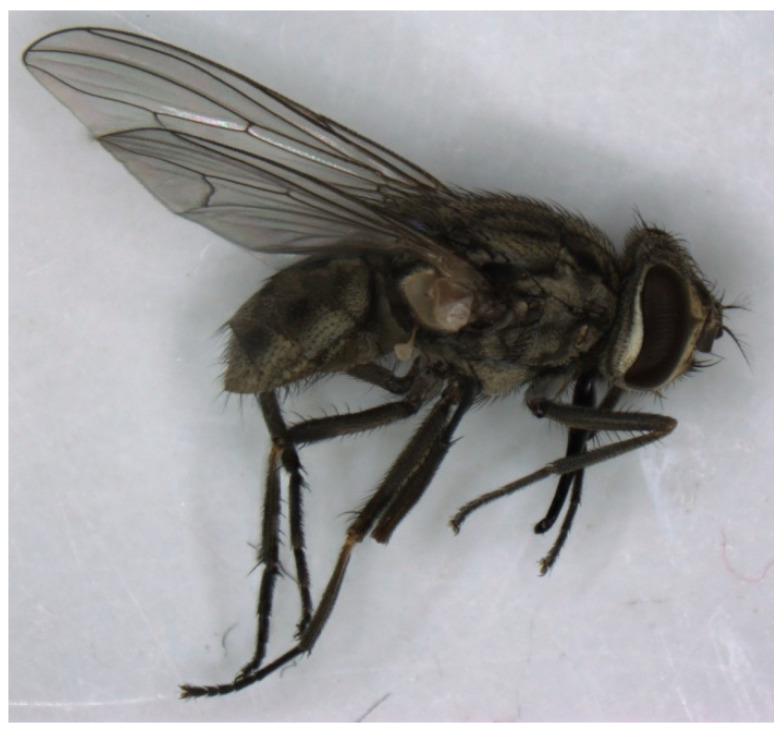
*Stomoxys calcitrans* (five to seven millimeter (mm) body length) in a lateral view.

**Figure 5 pathogens-12-00568-f005:**
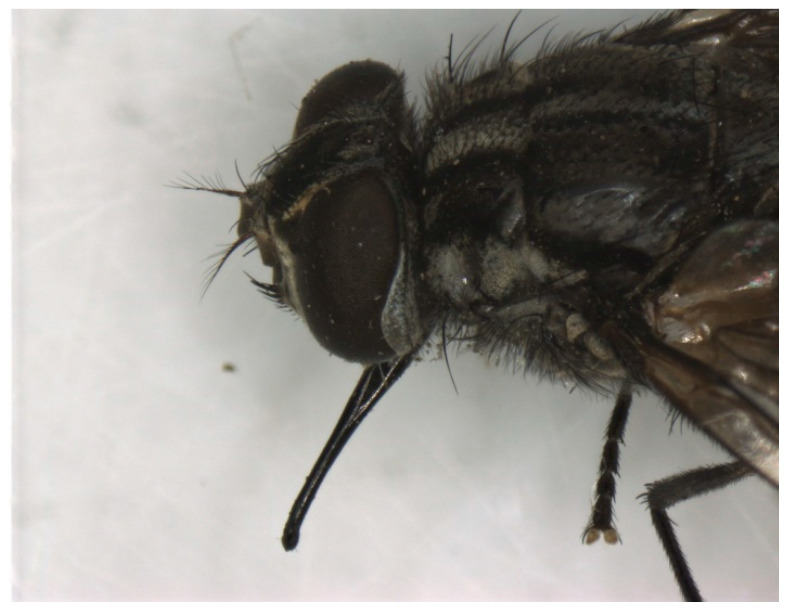
The head of *Stomoxys calcitrans* with the piercing and sucking mouthpart.

**Figure 6 pathogens-12-00568-f006:**
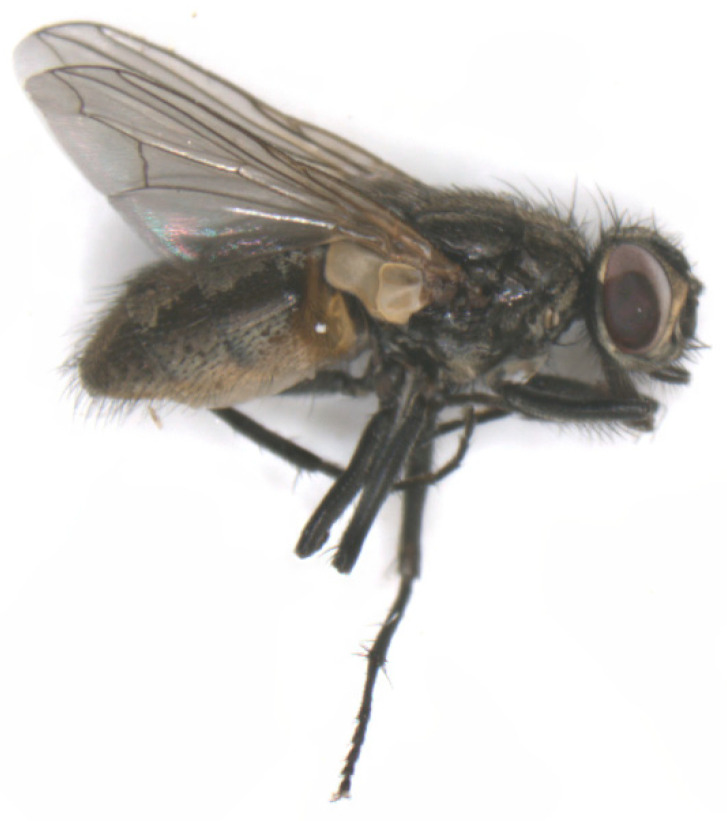
*Musca domestica* (six to seven mm body length) in a lateral view.

**Figure 7 pathogens-12-00568-f007:**
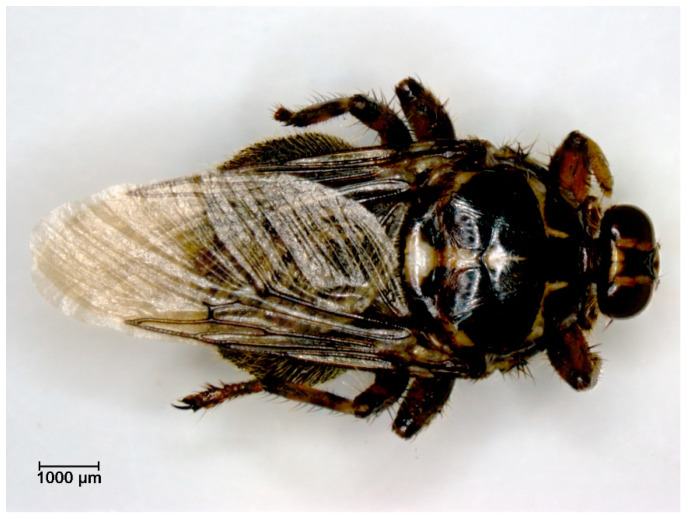
*Hippobosca equina* (seven to nine mm body length) in a dorsal view.

**Table 1 pathogens-12-00568-t001:** Incidence and prevalence of pathogens detected in the vectors.

Infectious Agent	Vector	Incidence	Prevalence	Location	Study
*Anaplasma* sp.	*Hippobosca equina*		3.8%	Cairo	Abdullah et al., 2022 [84]
*Borrelia* sp.	*Hippobosca equina*		2.9%	Cairo	Abdullah et al., 2022 [84]
*Corynebacterium pseudotuberculosis*	*Haematobia irritans*	2.4%		Northern California	Spier et al., 2004 [36]
*Corynebacterium pseudotuberculosis*	*Musca domestica* *Stomoxys calcitrans*	20%		Northern California	Spier et al., 2004 [36]
*Corynebacterium pseudotuberculosis*	*Musca domestica* *Stomoxys calcitrans*	19.3%		Northern California	Spier et al., 2004 [36]
*Corynebacterium pseudotuberculosis*	*Musca domestica*	0.3%		Northern California	Spier et al., 2004 [36]
*Cryptosporidium*	*Flies* (97.11% *Muscidae*)		50%	Northwest Georgia	Conn et al., 2007 [78]
*Habronema microstoma*	*Stomoxys calcitrans*		1.5–7.6% (estimated)	Teramo, central Italy	Traversa et al., 2008 [74]
*Habronema muscae*	*Musca domestica*		1.7–8.5% (estimated)	Teramo, central Italy	Traversa et al., 2008 [74]
*Habronema muscae*	*Musca domestica*		25.8%	Dubai	Schuster et al., 2013 [75]
*Habronema muscae*	*Musca domestica*		16.2%	Sharjah Emirate of the United Arab Emirates	Schuster et al., 2010 [51]
*Streptococcus equi* spp. *equi*	*Musca autumnalis*		0.54%	Central California	Pusterla et al., 2020 [73]
*Theileria equi*	*Stomoxys calcitrans*		3.2%	Hungary	Hornok et al., 2020 [79]

**Table 2 pathogens-12-00568-t002:** First concept: Equidae.

Equidae, Einhufer, Equiden, équidés
Horse, Pferd, cheval, chevaux
Donkey, Esel, âne

**Table 3 pathogens-12-00568-t003:** Second concept: relevant Brachycera for Equidae.

Order	Diptera Zweiflügler Diptères						
Suborder	Brachycera Fliege Brachycère						
Family	Glossinidae	Hippoboscidae louse flies Lausfliege hippoboscidés	Calliphoridae blow flies Schmeißfliege	Sarcophagida flesh fly Fleischfliege	Muscidae “echte Fliege”	Oestridae Botfly Dasselfliege Oestridés	Tabanidae Horsefly Bremse taon
Genus	*Glossina* *Tsetse fly* *Zungenfliege* *Tsetsefliege* *glossines*	*Hippobosca* *hippobosques* *Lipoptena*			*Muscinae*	*Oestrinae* *Nasendassel* *Hypodermatinae* *warble flies* *Hautdasseln* *Hypodermes* *Gasterophilinae* *Magendassel*	*Atylotus* *Chrysops* *Dasyrhamphis* *Glaucops* *Haematopota* *Heptatoma* *Hybomitra* *Nemorius* *Pangonius* *Philipomomyi* *Silvius* *Tabanus* *Therioplectes*
Species					*Hydrotaea* *Musca* *Stomoxys* *Haematobia* *Haematobosca*	*Cephenemyia* *Oestrus* *Pharyngomyia* *Rhinoestrus* *Hypoderma* *Oestromyia* *Portschinskia* *Bumblebee bot flies* *Przhevalskiana* *Gasterophilus*	

**Table 4 pathogens-12-00568-t004:** Questions for relevant data extraction.

What is the title of the study?
Who are the authors who contributed to the study?
In which year was the paper received, accepted and published?
In which journal was the paper published?
Which study design was used to collect and analyze the data?
What was the aim of the study?
Where was the study carried out (city, country, continent)?
What species are the host animals reported in the study?
Which methods are used to identify the vectors, the infectious agents and the transmission?
How did the authors identify the vectors morphologically?
Which species were detected as vectors or pests?
Which infectious agents transmitted by Brachycera were detected?
What is the prevalence of the vectors that tested positive?
Which is the prevalence of the equids that tested positive?
What were the pathological findings or clinical signs detected?
Are any prophylactic measures or therapies described to prevent or treat an infection or damage caused by Brachycera?
What were the results?
Can the transmission events be paired with a particular season?
Comments

## Data Availability

The list of data extracted, which are described in the text, from the 38 full texts to answer the questions relevant for data extraction presented in this study is available upon request from the corresponding author. The data are not publicly available due to the use of a licensing reference management program.

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
