# Peer review of "Relevant Brachycera (Excluding Oestroidea) for Horses in Veterinary Medicine: A Systematic Review"

_pathogens, 2023, doi:10.3390/pathogens12040568_

Round 1
Reviewer 1 Report
Nice study! I only have minor comments to help improve the manuscript.
Line 87: What is the meaning of ID?
Line 131: T.evansi --> T. evansi
Line 132: Trypanosoma spp. --> Trypanosoma spp.
Figures 2 and 3: Stomoxys calcitrans --> Stomoxys calcitrans
Figures 2, 3 and 4: Please add the scale bar.
Figure 4: Musca domestica --> Musca domestica
Line 336: nictating --> nictitating
Line 360: 2.2.2.11. Role of Muscidae in transmission of Theileria equi --> 2.2.2.11. Role of Muscidae in transmission of Theileria equi
Lines 579-580: Please check, H-P.F or HP.F
Author Response
Mag. med.vet Vicky Frisch
Clinical Unit of Equine Internal Medicine
University of Veterinary Medicine Vienna,
Veterinärplatz 1,
1210 Vienna, Austria
E-mail: Vicky.frisch@vetmeduni.ac.at
Dear Editor, Dear Reviewers,
At this point we wish to thank the reviewers for their helpful comments and suggestions. The manuscript has undergone a modification along the lines as suggested by the reviewers and the comments and suggestions have been incorporated into the revised manuscript.
Please find below the itemized list of changes and responses to each reviewers’ comments.
Reviewer 1:
Nice study! I only have minor comments to help improve the manuscript.
Response: We wish to thank the reviewer.
Line 87: What is the meaning of ID?
Response: unfortunately a copy paste error occurred. Has been deleted accordingly.
Line 131: T.evansi --> T. evansi
Response: Has been changed accordingly.
Line 132: Trypanosoma spp. --> Trypanosoma spp.
Response: Has been changed accordingly.
Figures 2 and 3: Stomoxys calcitrans --> Stomoxys calcitrans
Response: Has been changed accordingly.
Figures 2, 3 and 4: Please add the scale bar.
Response: Unfortunately, we are not in the possession of pictures of the flies (Stomoxys calcitrans, Musca domestica and Chrysops caecutiens) containing a scale bar. And as it is not allowed to include a scale bar retroactively, we added the body length of the different flies in the description of the particular pictures.
Figure 4: Musca domestica --> Musca domestica
Response: Has been changed accordingly.
Line 336: nictating --> nictitating
Response: Has been changed accordingly.
Line 360: 2.2.2.11. Role of Muscidae in transmission of Theileria equi --> 2.2.2.11. Role of Muscidae in transmission of Theileria equi
Response: Has been changed accordingly and later deleted due changes accordingly to the revisions of reviewer 2
Lines 579-580: Please check, H-P.F or HP.F
Response: Has been changed accordingly.
Author Response
Mag. med.vet Vicky Frisch
Clinical Unit of Equine Internal Medicine
University of Veterinary Medicine Vienna,
Veterinärplatz 1,
1210 Vienna, Austria
E-mail: Vicky.frisch@vetmeduni.ac.at
Dear Editor, Dear Reviewers,
At this point we wish to thank the reviewers for their helpful comments and suggestions. The manuscript has undergone a modification along the lines as suggested by the reviewers and the comments and suggestions have been incorporated into the revised manuscript.
Please find below the itemized list of changes and responses to each reviewers’ comments.
Reviewer 2:
General considerations
In this study, Frisch et al, propose a systematic review on the occurrence of equine pathogen in insects of the Brachycera suborder (Tabanidae, Muscidae, Glossinidae and Hippoboscidae). The systematic review that follows the PRISMA 2020 statement have allowed the selection of 38 manuscript that follow the criteria of inclusion as defined by the authors. This systematic study depicts that very little is known on the subject and that clearly additional studies are required to answer some basic questions raised by the systmetic literature survey. The methodology used in the study was:
1) to search databases using key words that specific for Equidae and translated in various languages combined with key words related to the Brachycera suborder and translates in diverse languages (English, German, French).
2) after duplicate elimination, to check for manuscript for inclusion criteria or exclusion criteria.
3) to extract data from the selected literature according to a defined questions that include those related to infectious agents among other.
4) to organize resulting data extraction into a systematic review report.
Response: We wish to thank the reviewer.
Suggestions for authors
The manuscript is clear in the statement of objectives and well written. Nevertheless, In the introductory section a better description of the Brachycera taxonomy will be valuable for the readers. Since paper focus on horse’s infectious agent transmitted by insects of the key the Brachycera suborder, definition and discussion on way pathogens are transmitted; In muscidae some members are passive transmitter of infectious agent, meaning not injecting pathogens but acting by regurgitation of defecation on the vertebrate hosts. Nevertheless, others are active transmitter (Stomoxes) acting by mechanical transmission. Biological amplification in the insects of the Brachycera suborder are also of veterinary interest (Glossinidae) that transmit trypanosomes after biological multiplication. All these concepts involved in the transmission of infectious agent must be carefully discussed in the introductory section.
Response: Has been changed accordingly to the reviewer’s suggestion.
The text in lines 33-36 now reads „ The order Diptera, uniting the “true flies”, is divided into two suborders, Nema-tocera and Brachycera, the latest being distinguished by shorter antennae [1]. The sub-order Brachycera includes amongst others Tabanidae, Muscidae, Glossinidae and Hippoboscidae, which are considered of great veterinary medical importance [1,2]. “.
The text in lines 58-60 now reads „ The transfer successfully occurs by regurgitation of infectious agents through their sponging mouthparts, defecation of the fly or while in mechanical contact with the host due to phoresis of different infectious agents [11,12].”
The text in lines 63-67 now reads „ The potential of Stomoxys spp. as mechanical vectors for infectious agents results from various adaptations of the flies to blood feeding including their piercing and sucking mouthparts. Transmission can potentially occur via regurgitation, defecation or phoresis of different infectious agents, traveling from one host to another via the flies´ body [6].”
The text in lines 74-79 now reads „ The flies play a role in transmission of the protozoan parasite, as they are blood feeders (anautogeny) and support cyclical development of the parasite. As soon as Trypanoso-ma congolense, Trypanosoma brucei spp. and Trypanosoma vivax enter the flies´body, they develop and start to multiply either in the midgut or proboscis [14]. By biting the host, the flies transfer the infective trypanosomes into the skin [14]. The bite of Glossinidae has been described as a sharp prick [13,15,16].”
Remarks:
- The manuscript is interesting but since not so much information can be extracted from the literature
some subheadings are very short when addressing pathogens species present in flies. Why do not
grouping pathogen in more genetic terms “Parasites, viruses, Bacteria”.
Response: Has been changed accordingly.
- Clearly, since city, country and continent belong to the data extracted why not providing a
comprehensive map about pathogens and Brachycera association for equine diseases? Likewise, why
exploiting all the data extracted during the literature survey? In a synthetic chapter that described all
the data extracted, for an historical view of research effort performed on this subject.
Response: A map showing the locations, where the studies were executed was added to provide more data.
- Why adding only pictures of Stomoxys and drosophila and not of Tabanidae or Hippobocidae?
Response: Has been changed accordingly, pictures of Chrysops sp. (Line 215) and Hippobosca equina (Line 430) were added.
- Why only performing research using English, German and French, knowing that may be most of report on south America might be in Spanish or Portuguese. This limits the opportunity to gather more
information on the topic.
Response: During the Literature search 15 texts were excluded because of the language barrier, as none of the authors is fluent in Spanish or Portuguese. Of those 15, 2 were written in Portuguese and 3 in Spanish. In those papers no major information, which would have changed the results, could be retracted from the English written abstracts.
Reviewer 3 Report
This manuscript is a review article covering some aspects of biting and secretion feeding Brachycera. It is not all encompassing and is less of a review than some relatively recent textbooks. However, as these flies see far less focus than Nematocera they are of some interest for entomologists and veterinarians to remind them of the possible relevance for disease and parasite/pathogen transmission.
I have several comments below.
I note that not all of the biting Brachycera are included in this review.
While this is somewhat a Nearctic phenomena it should be noted that the genus Symphoromyia is a significant biting pest of horses in many high elevation area. For example Symphoromyia atripes is a major biting pest of horses in Yellowstone National Park (Ross, 1940). This is not a unique problem it is often overlooked by entomologists that do not live in the western mountains of North America but as horse based outdoor recreation and horse based agricultural activities are significant there these flies might be worth noting. In regard to the often forgotten species and families you might at least also mention Suragina spp. in the Family Athericidae which are pests in certain locals again in North America. I am not sure if you also want to include the two eye exudate feeding Chloropidae in the genera Hippelates and Siphunculina. They transmit or can transmit several pathogens and directly cause disease and eye irritation.
Line 73: Hippobosca equina is a pest of horses in Asian and Africa as well as in Europe.
Following line 73: In addition, Lipoptena cervi a Hippobosicdae is an occasional pest of horses in Europe and at least one presenter in a not to old meeting discussed it as causing great distress in horses. See Haarløv 1964. I should note also that several species of Bartonella were detected in various Lipoptena.
Line 90-91: While this is true that younger horses have lower infection rates it is also a chronic lifelong disease so older horses have more time to simply be infected.
Line 102-103: Icosta americana is very unlikely to feed on a horse. This fly is primarily a specialist on large birds such as owls and hawks. I would suggest deleting this as it is not really relevant to the horse aspect of the manuscript.
Line 112: Bartonella are also likely transmitted by Hippoboscidae (Dehio et al) including a species that regularly attacks horses in Europe, Lipoptena cervi.
Line 116-117: While the literature on Q fever (caused by Coxiella burnetii) is quite extensive and often focuses on ticks or milk you might consider the paper by Nelder et al. where the infectious agent was discovered in Stomoxys calitrans (see below).
Line 116-117: Neorickettsia risticii is unlikely to be infecting a stable fly as they have no aquatic life stages to encounter snails.
Line 131: T.evansi needs a space.
Line 132. spp. should not be in italics.
Line 139: I think you need to state this is a helminth somewhere in this line not just a disease.
Line 150: Same as above. Not just parasitic but a helminth.
Line 156: see above same issue.
Line 224: I believe you need to cite the series of experiments here.
Line 251: Cite here. This is a dubious clam in the publication as it is very hard to replicate with Culicoides being the almost certain vectors.
Line 449-451: From what I have seen very few studies of Stomoxys indicate it is a real vector of protozoans and instead is quiet refractory to transmission.
DEHIO, C. U. SAUDER, AND R. HIESTAND. 2004. Isolation of Bartonella schoenbuchensis from Lipoptena cervi, a blood-sucking arthropod causing deer ked dermatitis. Journal of Clinical Microbiology 42: 5320–5323.
Haarløv, Niels. “Life Cycle and Distribution Pattern of Lipoptena Cervi (L.) (Dipt., Hippobosc.) on Danish Deer.” Oikos, vol. 15, no. 1, 1964, pp. 93–129.
Nelder, M.P., J.E. Lloyd, A.D. Loftis, and W.K. Reeves. 2008. Coxiella burnetii in wild-caught filth flies. Emerging Infectious Diseases 14: 1002-1004.
Ross. H.H. 1940. The Rocky Mountain Black Fly, Symphoromyia atripes (Diptera : Rhagionidae). Annals of the Entomological Society of America 33(2): 254-257.
Author Response
Mag. med.vet Vicky Frisch
Clinical Unit of Equine Internal Medicine
University of Veterinary Medicine Vienna,
Veterinärplatz 1,
1210 Vienna, Austria
E-mail: Vicky.frisch@vetmeduni.ac.at
Dear Editor, Dear Reviewers,
At this point we wish to thank the reviewers for their helpful comments and suggestions. The manuscript has undergone a modification along the lines as suggested by the reviewers and the comments and suggestions have been incorporated into the revised manuscript.
Please find below the itemized list of changes and responses to each reviewers’ comments.
Reviewer 3:
This manuscript is a review article covering some aspects of biting and secretion feeding Brachycera. It is not all encompassing and is less of a review than some relatively recent textbooks. However, as these flies see far less focus than Nematocera they are of some interest for entomologists and veterinarians to remind them of the possible relevance for disease and parasite/pathogen transmission.
I have several comments below.
Response: We wish to thank the reviewer.
I note that not all of the biting Brachycera are included in this review.
While this is somewhat a Nearctic phenomena it should be noted that the genus Symphoromyia is a significant biting pest of horses in many high elevation area. For example Symphoromyia atripes is a major biting pest of horses in Yellowstone National Park (Ross, 1940). This is not a unique problem it is often overlooked by entomologists that do not live in the western mountains of North America but as horse based outdoor recreation and horse based agricultural activities are significant there these flies might be worth noting. In regard to the often forgotten species and families you might at least also mention Suragina spp. in the Family Athericidae which are pests in certain locals again in North America. I am not sure if you also want to include the two eye exudate feeding Chloropidae in the genera Hippelates and Siphunculina. They transmit or can transmit several pathogens and directly cause disease and eye irritation.
Response: During my research in the different engines, I was not able to find the paper written by Ross H.H., thank you very much for providing this knowledge. I immediately included it in the introduction. I also included Suragina spp. (Stuckenberg, B.R. 2000) as this information was not assessed during the literature research, as well as Chloropidae (Machtinger et al., Pests and parasites of horses).
The text in lines 185-192 now reads „ Symphoromyia spp. of the family Rhagionidae, a major biting pest of horses in Yellow-stone National Park, were described attacking horses severely in many high elevation area [57]. Additionally, other flies like Suragina spp. as well as the exudate feeding Chloropidae in the genera Hippelates, Liohippelates and Siphunculina are discussed pests of horses [58,59]. While rasping the skin with spines on their legs to increase secretion production of the host´s skin, Chloropidae (eye gnats) are also described to feed on equids beside S. calcitrans. Irritations of the eyes as well as the possibility of transmission of different bacterial and viral agents are linked to these flies [59].”
Line 73: Hippobosca equina is a pest of horses in Asian and Africa as well as in Europe.
Response: My wording was clearly misleading but as new information were added, the geographical location was deleted.
The text in lines 83-85 now reads, „ Hippoboscidae, also known as louse flies, are bloodsucking flies. Hippobosca equina is the most common specie to parasitize Equidae[15,18]. Additionally horses are dis-cussed to be a facultative host of Lipoptena cervi [19].”
Following line 73: In addition, Lipoptena cervi a Hippobosicdae is an occasional pest of horses in Europe and at least one presenter in a not to old meeting discussed it as causing great distress in horses. See Haarløv 1964. I should note also that several species of Bartonella were detected in various Lipoptena.
The text in lines 83-85 now reads, „ Additionally horses are dis-cussed to be a facultative host of Lipoptena cervi [19].”
Line 112: Bartonella are also likely transmitted by Hippoboscidae (Dehio et al) including a species that regularly attacks horses in Europe, Lipoptena cervi.
Response: The manuscript has been changed accordingly to the two proposals. Information from two studies which were not extracted during the literature search (Dehio et al. 2004 and Pena-Espinoza et al. in review) were added.
The text in lines 143-144 now reads, „ Hippobosca equina and L. cervi are considered potential vectors for bacterial agents such as Bartonella spp. [18,19].”
Line 90-91: While this is true that younger horses have lower infection rates it is also a chronic lifelong disease so older horses have more time to simply be infected.
Response: Has been changed accordingly.
The text in lines 101-103 now reads, „ Younger horses are apparently more active in shooing away Tabanids, which could be one explanation for the higher infection rates of EIAV in adult horses [2]. Nevertheless, EIA is a lifelong disease and older equids have more time to simply be infected.”
Line 102-103: Icosta americana is very unlikely to feed on a horse. This fly is primarily a specialist on large birds such as owls and hawks. I would suggest deleting this as it is not really relevant to the horse aspect of the manuscript.
Response: We added the information of Icosta americana being ornithophilic (Line 115). As it is a possibility of xeno-diagnostic for WNV, which is considered an important virus in equine medicine, we discussed this matter and concluded to not deleting the information.
Line 116-117: While the literature on Q fever (caused by Coxiella burnetii) is quite extensive and often focuses on ticks or milk you might consider the paper by Nelder et al. where the infectious agent was discovered in Stomoxys calitrans (see below).
Response: Has been changed accordingly and added in Line 127.
Line 116-117: Neorickettsia risticii is unlikely to be infecting a stable fly as they have no aquatic life stages to encounter snails.
Response: Has been changed accordingly.
The text in lines 128-131 now reads, „ Bartonella spp. are also suspected of being transmitted via stable flies [33]. It was demonstrated that S. calcitrans can harbour Neorickettsia ristricii but no attempt of transmission was successful [35].”
Line 131: T.evansi needs a space.
Response: Has been changed accordingly.
Line 132. spp. should not be in italics.
Response: Has been changed accordingly.
Line 139: I think you need to state this is a helminth somewhere in this line not just a disease.
Response: Has been changed accordingly to “helminth infection” (Line 156)
Line 150: Same as above. Not just parasitic but a helminth.
Response: Has been changed accordingly to “helminth infection” (Line 167)
Line 156: see above same issue.
Response: Has been changed accordingly to “helminthic disease” (Line 173)
Line 224: I believe you need to cite the series of experiments here.
Response: My wording was clearly misleading, it is only one study including comparisons. Has been changed accordingly.
Line 251: Cite here. This is a dubious clam in the publication as it is very hard to replicate with Culicoides being the almost certain vectors.
Response: Has been changed accordingly and discussed in the discussion.
The text in lines 285-286 now reads, „ However, it remains unclear very unlikely if S. calcitrans is a competent vector for this pathogen.”
Line 449-451: From what I have seen very few studies of Stomoxys indicate it is a real vector of protozoans and instead is quiet refractory to transmission.
Response: Has been changed accordingly, the wording phoresis was included to provide a better description.
The text in lines 365-371 now reads, „ Oocysts of Cryptosporidium parvum and Giardia lamblia were detected on the surface of Muscidae (not further defined) suggesting phoresis, possibly leading to me-chanical transmission by oocytes sticking onto the mouthparts and tarsal adhesive structures of the flies. Further, both parasites were detected in fly homogenates ana-lysed using fluorescent in situ hybridization (FISH) and immunofluorescent antibody (IFA) assays, showing a contamination inside the fly bodies.”
DEHIO, C. U. SAUDER, AND R. HIESTAND. 2004. Isolation of Bartonella schoenbuchensis from Lipoptena cervi, a blood-sucking arthropod causing deer ked dermatitis. Journal of Clinical Microbiology 42: 5320–5323.
Haarløv, Niels. “Life Cycle and Distribution Pattern of Lipoptena Cervi (L.) (Dipt., Hippobosc.) on Danish Deer.” Oikos, vol. 15, no. 1, 1964, pp. 93–129.
Nelder, M.P., J.E. Lloyd, A.D. Loftis, and W.K. Reeves. 2008. Coxiella burnetii in wild-caught filth flies. Emerging Infectious Diseases 14: 1002-1004.
Ross. H.H. 1940. The Rocky Mountain Black Fly, Symphoromyia atripes (Diptera : Rhagionidae). Annals of the Entomological Society of America 33(2): 254-257.
Response: We wish to thank the reviewer for the helpful references.
Round 2
Reviewer 2 Report
Thank you for considering my remarks.
Author Response
We wish to thank the reviewer for his positive evaluation of our revised manuscript. The manuscript was sent for professional English language proof reading prior to submission.